# Learning Dual Enhanced Representation for Contrastive Multi-view Clustering

## ABSTRACT

Contrastive multi-view clustering is widely recognized for its effectiveness in mining feature representation across views via contrastive learning (CL), gaining significant attention in recent years. Most existing methods mainly focus on the feature-level or/and cluster-level CL, but there are still two shortcomings. Firstly, feature-level CL is limited by the influence of anomalies and large noise data, resulting in insufficient mining of discriminative feature representation. Secondly, cluster-level CL lacks the guidance of global information and is always restricted by the local diversity information. We in this paper Learn dUal enhanCed rEpresentation for Contrastive Multi-view Clustering (LUCE-CMC) to effectively addresses the above challenges, and it mainly contains two parts, i.e., enhanced feature-level CL (En-FeaCL) and enhanced cluster-level CL (En-CluCL). Specifically, we first adopt a shared encoder to learn shared feature representations between multiple views and then obtain cluster-relevant information that is beneficial to the clustering results. Moreover, we design a reconstitution approach to force the model to concentrate on learning features that are critical to reconstructing the input data, reducing the impact of noisy data and maximizing the sufficient discriminative information of different views in helping the En-FeaCL part. Finally, instead of contrasting the view-specific clustering result like most existing methods do, we in the En-CluCL part make the information at the cluster-level more richer by contrasting the cluster assignment from each view and the cluster assignment obtained from the shared fused features. The end-to-end training methods of the proposed model are mutually reinforcing and beneficial. Extensive experiments conducted on multi-view datasets show that the proposed LUCE-CMC outperforms established baselines to a considerable extent.

## CCS CONCEPTS

• **Computing methodologies** → **Cluster analysis**.

## KEYWORDS

Deep multi-view clustering, contrastive learning, representation learning

## 1 INTRODUCTION

In the era of Industry 4.0 and the global spread of big data, multi-view clustering (MVC) [8, 10–12] has received extreme attention

*ACM MM, 2024, Melbourne, Australia*

© 2024 Copyright held by the owner/author(s). Publication rights licensed to ACM.
ACM ISBN 978-x-xxxx-xxxx-x/YY/MM
https://doi.org/10.1145/nnnnnnn.nnnnnnn

from scholars due to its ability to capture the relationships among different views or modals of the same object. Existing MVC methods have made important breakthroughs in fields such as 3D object recognition [16], pedestrian captioning [17] and object detection [6].

Contrastive MVC [23, 26, 30, 33] methods have gained lots of attention due to its excellent performance in clustering via contrastive learning (CL). It can not only effectively solve the challenges of multi-view clustering, but also ensure that global information can be explored without destroying the internal structure of the data. Xue et al. [30] proposed to use contrastive learning to regularize intra- and inter-data correlation that is beneficial to the clustering task to the greatest extent. However, this approach ignores features that only appear in a specific view and contribute to the clustering task. Wang et al. [23] utilized the relationships in each view to eliminate noise in the data by selecting positive and negative samples for graph CL on a unified graph. This method uses graph CL to learn public feature representations to the greatest extent, but CL is susceptible to the influence of abnormal data, and lacks global information guidance in the cluster-level CL process, reducing the diversity of perspective information. With the help of deep learning and end-to-end training, contrastive MVC performance has been significantly improved. The most existing contrastive MVC methods mainly focus on feature-level and cluster-level CL, but there are still two challenges.

The first is that feature-level CL is sensitive to anomalies and high levels of noise. These anomalies and noise data have a particular impact on the feature space because they may distort the true distribution of the data, resulting in insufficient mining of discriminative feature representation. This lowers the upper limit of mining useful information for clustering tasks between views. The second is that cluster-level CL lacks consideration of global information. Because cluster-level CL often focuses on local features or differences between relatively independent data clusters, models may not fully capture and exploit global structure and information across different views or sources. The limitations of this method may result in the diversity and richness of information not being fully exploited, which in turn affects the final clustering quality.

To address the above challenges, we Learn dUal enhanCed rEpresentation for Contrastive Multi-view Clustering (LUCE-CMC). LUCE-CMC lies in two critical enhancements: enhanced feature-level contrastive learning (En-FeaCL) and enhanced cluster-level contrastive learning (En-CluCL). The dual enhanced representation manifests itself in En-FeaCL through the augmentation of latent feature representation via feature alignment and reconstitution. While En-CluCL achieves an enhanced clustering structure through the utilization of a shared representation. First, our method adopts a shared encoder, aiming to maximize the extraction of shared feature representations in multi-view data, laying the foundation for a comprehensive understanding of the inherent characteristics of the data.

Subsequently, feature alignment is implemented to enhance the similarity between different data samples, thereby significantly enhancing the model's robustness in processing the input data. Nonetheless, feature alignment is somewhat limited by view-specific information. To mitigate this, we employ autoencoders, leveraging their unique reconstruction mechanism to force the model to prioritize learned features necessary for accurate reconstruction of the input data. This method can effectively remove view-specific features, thereby promoting En-FeaCL to learn sufficient discriminative feature representations. Furthermore, unlike previous cluster-level CL methods, our method performs contrastive learning by contrasting the cluster assignments of different views with the assignments obtained from the fused features. Throughout the end-to-end learning process, all component modules of the LUCE-CMC framework are collaboratively optimized to ensure enhanced collaboration and ultimately superior clustering results. This dual enhanced representation method not only ensures the robustness and adaptability of the model when dealing with complex data, but also emphasizes the importance of sufficient discriminative features.

The main contributions are as follows:

- We propose a novel LUCE-CMC method, which effectively mines feature representations related to clustering tasks and removes redundant information harmful to clustering tasks in an end-to-end manner.
- A new enhanced feature-level CL module has been designed, which combines the concepts of feature alignment and reconstitution to force the model to focus on learning features that are critical for reconstructing the input data. By doing so, it effectively mitigates the impact of noisy data and maximizes the learning of sufficient discriminative information.
- A novel enhanced cluster-level CL mechanism is proposed to enhance cluster representation and increase the diversity of contrastive samples and make full use of the fused high-level features to better capture the similarities and differences between data.
- Experimental evaluations conducted on various challenging multi-view datasets demonstrate the superior performance of the proposed approach over several traditional MVC methods and state-of-the-art deep MVC methods.

## 2 RELATED WORK

In this part, we introduce the related works on contrastive learning and deep multi-view clustering.

### 2.1 Contrastive Learning

In the field of self-supervised learning, contrastive learning has received increasing attention, playing an important role in promoting the advancements across a myriad of application areas. This approach distinguishes itself by leveraging the intrinsic structure of data without reliance on explicit labels, facilitating a deeper understanding and representation of data characteristics. Chen et al. [3] pioneered a method wherein an augmented version of the original datum serves as the positive sample, while all other data points within the dataset are treated as negative samples, thereby enhancing the discriminative capacity of the model. By extending

the utility of contrastive learning, Hu et al. [7] synergistically combined it with a robust Gaussian mixture model, thereby augmenting the precision and efficacy of text clustering methodologies. Xu et al. [29] worked by integrating instance-level contrastive learning with multi-scale graph convolutional networks, significantly enriching the depth and accuracy of clustering tasks pertinent to image data. This evolution underscores the transformative potential of contrastive learning in the self-supervised learning, fostering a more comprehensive exploitation of unlabeled data across various domains.

### 2.2 Deep Multi-view Clustering

Deep multi-view clustering uses the powerful feature extraction and representation learning capabilities of deep learning models to effectively reduce the dimensionality of data, while filtering out noise and improving the accuracy and robustness of clustering. On the one hand, deep MVC is superior to traditional multi-view clustering because it uses a deep neural network to more effectively mine and explore potential information between different views, and at the same time can solve high-dimensional data problems. On the other hand, it is better than single-view clustering, because the single-view object information described by the data is very limited and ignores additional important information.

To name a few, Chen et al. [4] ingeniously integrated the principles of federated learning with the objective of extracting complementary clustering architectures from a group of clients. This innovative approach not only facilitates the identification of synergistic cluster configurations across diverse data sources but also adeptly addresses the prevailing challenges associated with data incompleteness and the safeguarding of privacy. By expanding upon the paradigm of multi-dimensional data analysis, Wang et al. [24] architect a triadic rank contrastive learning framework. This sophisticated model is predicated on the exhaustive elicitation of salient information across three hierarchical layers. By doing so, it systematically uncovers and leverages intricate patterns within data, thereby significantly improving the effectiveness of information retrieval and data interpretation processes. By contributing to the advancement of adaptive learning methodologies, Wang et al. [25] introduced a new deep sparse regularizer learning mechanism. This model distinguishes itself by its capacity to autonomously derive data-driven sparse regularizers, thereby optimizing the representation and processing of data.

## 3 THE PROPOSED METHOD

### 3.1 Problem Formulation

Consider a multi-view learning scenario with $m$ discrete random variables $\{X^1, X^2, \ldots, X^m\}$ observed from $m$ different input views. Each data sample $X^i = \{x_1^i, x_2^i, \ldots, x_n^i\} \in \mathbb{R}^{n \times d^i}$ consists of $n$ observations in the $i$-th view, where $d^i$ is the feature dimension, and $n$ is the number of data samples in each view. Each observation $x_j^i \in \mathbb{R}^{d^i}$ is a vector with $d^i$ dimensions. Each view is treated as a discrete random variable $X^i$, denoted by the index $i$. The aim of LUCE-CMC is to learn dual enhanced representation for feature-level CL improvement with alignment and reconstitution, and cluster-level CL improvement using shared representation. During the end-to-end

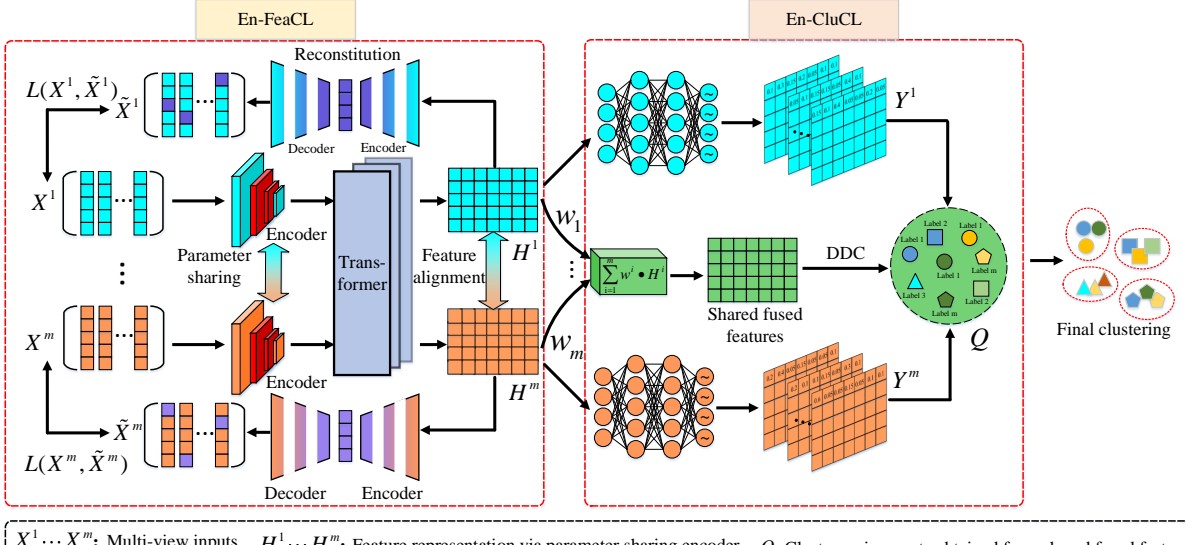

**Figure 1: The framework of LUCE-CMC method. First, with the input multi-view data $\{X^i\}_{i=1}^m$, the feature representation of each view $\{H^i\}_{i=1}^m$ is obtained through parameter sharing encoder, then enhanced feature-level CL (En-FeaCL) through feature alignment and reconstitution method are imposed. Afterwards, the feature representation $\{H^i\}_{i=1}^m$ feeds into En-CluCL mechanism, which contrasts the cluster assignments obtained from fused features with the assignments from each view. Thus the global information is compressed to the greatest extent, promoting complementarity and diversity at the cluster-level. Finally more satisfactory clustering results are obtained.**

learning process, all modules within the LUCE-CMC framework are optimized collaboratively to ensure improved cooperation and superior clustering outcomes.

## 3.2 Overall Framework

Figure 1 shows the overall framework of our proposed model, including enhanced feature-level CL (En-FeaCL), and cluster-level CL (En-CluCL) mechanism, and deep divergence-based clustering (DDC) module. En-FeaCL mechanism consists of feature contrastive alignment and reconstitution module. The feature contrastive alignment and reconstitution module ensure global information exploration without compromising internal data structure. En-CluCL mechanism conducts contrastive learning between cluster assignments obtained from the fused features with assignment from each view, leveraging rich fused features for increased diversity and improved contrast performance. DDC module utilizes clustering space structure to enhance the compactness of similar samples and separability of dissimilar samples during optimization.

## 3.3 Enhanced Feature-level Contrastive Learning Mechanism

By observing that most existing feature-level CL based methods are sensitive to noise, we design an enhanced feature-level contrastive learning (En-FeaCL) mechanism, which includes feature alignment and reconstruction model.

Feature alignment can more effectively explore global information without destroying the internal structure of the data. When

facing multi-view data input, feature alignment can enhance the similarity between different samples and improve the robustness of the model to the input data. By using contrastive learning to achieve feature alignment, the model can be prompted to learn richer and more representative feature representations that better capture the intrinsic structure and similarities of the data. Therefore the loss function is

$$\mathcal{L}_{ali} = \frac{1}{n} \sum_{i=1}^n \frac{1}{\binom{m}{2}} \sum_{u=1}^m \sum_{v=u+1}^m \mathbb{1}_{u \neq v} l_i^{uv}, \qquad (1)$$

where $\mathbb{1}_{u \neq v}$ takes the value of 1 when the constraint $u \neq v$ is satisfied. $l_i^{uv}$ is defined as

$$l_i^{uv} = -\mathbb{E}\left[ \log \frac{e^{s_{ii}^{uv}(h_i^u, h_i^v)/\tau_1}}{\sum_{s' \in Negative\_sample(h_i^u, h_i^v)} e^{s'/\tau_1}} \right], \qquad (2)$$

where $\tau_1$ denotes the temperature hyperparameter of feature alignment, and $Negative\_sample(h_i^u, h_i^v)$ indicates the collection of all negative sample pair similarities. The cosine similarity $s_{ij}^{uv}(h_i^u, h_j^v)$ of two feature representations $h_i^u$ and $h_j^v$ refers to

$$s_{ij}^{uv}(h_i^u, h_j^v) = \frac{(h_i^u)^T h_j^v}{\|h_i^u\| \cdot \|h_j^v\|}. \qquad (3)$$

However, in the process of feature alignment, a major challenge we face is how to effectively eliminate or minimize view-specific information in the data. This often prevents the model from learning common feature representations with higher generalization

ability from different views. The autoencoder uses its unique reconstruction mechanism to force the model to concentrate on learning those features that are crucial to reconstructing the input data (i.e., feature representation $\{H^i\}_{i=1}^m$). This is crucial to achieve high-quality feature alignment, as the success of feature alignment relies heavily on the ability to effectively express features from different data sources. On the one hand, it ignores those features that only appear in specific viewpoints and has limited contribution to the clustering task. On the other hand, it is possible to automatically extract useful and compact feature representations from the data. In short, reconstruction mechanism not only improves the quality of feature representation, but also promotes the optimization of the feature alignment process, which is beneficial to enhance the learning of more potential representations of multi-view data.

We use mean square error (MSE) loss to implement the reconstitution idea. The loss function is:

$$\mathcal{L}_{recon} = \frac{1}{m} \sum_{i=1}^m ||\mathbf{X}^i - \tilde{\mathbf{X}}^i||^2, \tag{4}$$

where $X^i$ represents the original input, $\tilde{X}^i = \mu(H^i)$ represents the predicted value obtained by reconstructing the feature representation $H^i$ through the autoencoder network $\mu(\cdot)$.

The loss of enhanced feature-level contrastive learning function is:

$$\mathcal{L}_{En-FeaCL} = \mathcal{L}_{ali} + \mathcal{L}_{recon}. \tag{5}$$

Unlike Shu et al. [20], our method involves first deriving the feature representation $\{H^i\}_{i=1}^m$ via an encoder from $\{X^i\}_{i=1}^m$, followed by feeding $\{H^i\}_{i=1}^m$ into an autoencoder to reconstruct the data $\{\tilde{X}^i\}_{i=1}^m$. In contrast, Shu et al. [20] feed the original data $\{X^i\}_{i=1}^m$ directly into the autoencoder. This method is limited by the complex, high-dimensional nature of multi-view data, hindering the model's ability to discern inherent data patterns. Consequently, our method's feature representation $\{H^i\}_{i=1}^m$ effectively captures the data's deep features and essential attributes. Additionally, this ensures that $\{H^i\}_{i=1}^m$ retains sufficient information for data reconstruction, indirectly validating the effectiveness and completeness of our feature extraction model.

## 3.4 Enhanced Cluster-level Contrastive Learning Mechanism

In existing multi-view contrastive learning methods, cluster-level contrastive learning (CluCL) based MVC methods have gained lots of attention. Cluster assignments are obtained through multi-layer perceptron networks and activation functions, and cluster assignments between different views are contrasted and learned. It can effectively improve the distinction of learned features. But existing methods heavily depend on the quality of clustering. If the clustering results are inaccurate, it may lead to learning wrong or inaccurate feature representations. With this as motivation, we propose an Enhance cluster-level CL (En-CluCL) mechanism. We further enhance the effectiveness of feature representation by contrasting the cluster assignments between different views with that of the fused features. With the fused features as the dominant ones, cluster-level CL are more comprehensive and efficient, mitigating the impact of single-view bias. Compared with previous methods,

the En-CluCL module can capture more comprehensive data features and improve the complementarity at the clustering level. This complementarity helps to improve the accuracy of clustering and the richness of feature representation.

Specifically, we contrast the obtained perspective features $\{Y^i\}_{i=1}^m$ with the fused features $Q$. Our method introduces an additional representation space during the mapping process. This representation space allows us to compare label-level and fused high-level features. We can use Eq. (3) to get the cosine similarity $s_{ij}^u(q_i, y_j^u)$ between two features in $Q$ and $\{Y^i\}_{i=1}^m$. The loss function is written as

$$\mathcal{L}_{En-CluCL} = \frac{1}{2} \sum_{u=1}^m \mathbb{1}_{u \neq v} l^u - Y(Q), \tag{6}$$

where $Y(Q)$ represents the entropy of the cluster assignment probabilities, $Y(Q) = \sum_{i=1}^m \sum_{j=1}^c (\frac{1}{N} \sum_{k=1}^N y_{kj}^i) \log(\frac{1}{N} \sum_{k=1}^N y_{kj}^i)$ calculates the entropy of the fused feature $Q$. The first term in Eq. (6) is responsible for learning the consistency across clusters from different modalities, while the second term is used for regularization, in order to avoid trivial solutions. $l^u$ is defined as

$$l^u = -\frac{1}{r} \sum_{i=1}^r \log \frac{e^{s(q_i, y_i^u)/\tau_2}}{\sum_{j=1}^r (e^{s(q_i, y_j^u)/\tau_2})}, \tag{7}$$

where $\tau_2$ represents the temperature hyperparameter of the En-CluCL mechanism.

## 3.5 Deep Divergence-based Clustering Module

Since the K-Means clustering algorithm can only handle data with a single perspective and cannot effectively fit nonlinear data, we use the deep divergence-based clustering (DDC) method. The DDC divergence-based metric is used as the clustering criterion, by maximizing divergence principle is used to measure the differences between data distributions to achieve more accurate data clustering.

DDC losses consist of three main components. The first term is a generalized version of Cauchy-Schwartz divergence, which measures the divergence between cluster centers and data distributions. The second term ensures that the vectors of clustering results are orthogonal. Finally, the third term introduces geometric structure into the Cauchy-Schwartz divergence, helping to prevent trivial solutions from being learned. By fusing these three terms, DDC loss can effectively optimize the clustering results of multi-view nonlinear data while taking into account the distribution differences between them. The overall function of DDC loss is given by

$$\mathcal{L}_{DDC} = \frac{1}{k} \sum_{i=1}^{k-1} \sum_{j>i} \frac{\mu_i^T \mathbf{E} \mu_j}{\sqrt{\mu_i^T \mathbf{E} \mu_i \mu_j^T \mathbf{E} \mu_j}} + triu(C^T C)$$

$$+ \frac{1}{k} \sum_{i=1}^{k-1} \sum_{j>i} \frac{\gamma_i^T \mathbf{E} \gamma_j}{\sqrt{\gamma_i^T \mathbf{E} \gamma_i \gamma_j^T \mathbf{E} \gamma_j}}. \tag{8}$$

The equation calculates the DDC loss using various terms. Here, $k$ is the total count of clusters, while $\mathbf{E}$ signifies a matrix derived using the Gaussian kernel. The term $\mu_i$ is the column vector from the clustering outcome $C$. Furthermore, $\gamma_i$ is identified as the $i$-th

---

**Algorithm 1** The proposed algorithm

**Input**: Multi-view dataset $\{X^i\}_{i=1}^m$; number of clusters $c$
**Parameter**: Hyperparameters $\alpha$, $\tau_1$, $\tau_2$; learning rate $\gamma$
**Output**: The label predictor $C$

1: Initialize the neural network parameters $\{\theta^i\}_{i=1}^m$.
2: **while** not converge **do**
3:     Extract view-specific representations $\{H^i\}_{i=1}^m$ by sharing view-specific encoders.
4:     Enhanced feature-level contrastive learning:
5:         Compute the feature contrastive alignment loss Eq. (1).
6:         Compute the reconstitution loss Eq. (4).
7:         Compute the En-FeaCL loss Eq. (5).
8:     Compute the En-CluCL loss Eq. (6).
9:     Compute the DDC loss by Eq. (8).
10:     Optimize the overall loss Eq. (9) by adam optimizer and back-propagation.
11: **end while**
12: **return** $C$

---

column vector from the matrix $U_{ab} = exp(- \parallel \alpha_a - e_b \parallel^2)$, with $e_b$ being the simplex's $b$-th vertex. The expression $triu(C^T C)$ denotes the summation of the elements in the upper triangular portion of the matrix.

## 3.6 Optimization

The models we proposed promote each other in an end-to-end training manner and achieve mutual benefit to achieve our goals. By jointly optimizing the model, satisfactory clustering results are obtained. The overall loss of this model is

$$\mathcal{L}_{total} = \mathcal{L}_{DDC} + \alpha \mathcal{L}_{En-FeaCL} + (1 - \alpha)\mathcal{L}_{En-CluCL}. \quad (9)$$

where $\alpha$ is the trade-off parameter of contrastive learning in the overall loss. Algorithm 1 provides a comprehensive overview of the proposed framework, offering a detailed and structured representation of the computational steps involved in the algorithmic process.

## 3.7 Connection with Information Bottleneck Theory

Information bottleneck methods [9, 31] aim to identify key data features by minimizing the compression information and maximizing the preserved information. Yan et al. [32] applied the information bottleneck concept to multimodal clustering, treating it as a two-stage data compression process across heterogeneous modalities. The information bottleneck methodology is based on two key concepts: "compression" and "preservation".

"Compression" involves reducing the information that represents input data. This process aims to extract a refined data representation with reduced information content, yet retaining the original data's key features. Our En-FeaCL mechanism leverages view-specific encoders and autoencoders for reconstructing and extracting global features from each view. This method transforms raw and high-dimensional data into a compact and low-dimensional feature space. This process aligns with the information bottleneck theory, aiming

to remove redundant information from a multi-view perspective. It also ensures the preservation of critical information, enhancing data efficacy and relevance for downstream tasks.

"Preservation" in data compression ensures that information crucial to output variables or task goals is retained. This principle states that the compressed representation must retain enough information to enable accurate predictions, like clustering outcomes, despite data compression. Our En-CluCL mechanism enhances the accuracy of cluster-level contrastive learning. This enhancement comes from comparing cluster assignments across different views with fused features. En-CluCL optimizes the retention of key features in fused representations, enriching the information content vital for clustering tasks. Essentially, En-CluCL upholds the information preservation principle, satisfying data compression needs while maintaining the accuracy and effectiveness of clustering outcomes.

## 4 EXPERIMENTS

### 4.1 Datasets

We evaluate our proposed method on five challenging multi-view datasets. **Caltech** [5] dataset contains 1440 samples from seven clusters. The dataset provides five feature processing methods, namely WM, CENTRIST, LBP, GIST and HOG. We consider each of these features as a modality with increasing viewing angle to evaluate the robustness of the proposed method. **CCV** [14] dataset contains 6773 video samples and 20 types of classes. The three views of this dataset are SIFT, STIP, and MFCC. The details of the dataset are shown in Table 1.

### 4.2 Evaluation Metrics

In our experiments, we employ two widely recognized metrics: accuracy (ACC) and normalized mutual information (NMI). ACC measures the consistency of prediction results with real labels, while NMI assesses the similarity between clustering results and real labels. Higher values for both metrics indicate better model performance.

### 4.3 Baselines

In order to further verify the superiority of the proposed method, we used a total of eighteen baseline methods for comparative experiments, including two classic single-view clustering methods (**KM** (K-Means), **Ncuts** (Normalized Cuts)), two full single-view clustering methods (**AvKM** (All-view K-Means), **AvNcuts**

**Table 1: Details about the multi-view datasets**

| Dataset | Type | Views | Dimensionality |
|---|---|---|---|
| Caltech-2V | image | 2 | 40, 254 |
| Caltech-3V | image | 3 | 40, 254, 928 |
| Caltech-4V | image | 4 | 40, 254, 928, 512 |
| Caltech-5V | image | 5 | 40, 254, 928, 512, 1984 |
| CCV | video | 3 | 5000, 5000, 4000 |

**Table 2: Clustering results in terms of ACC and NMI on the multi-view datasets.**

| Methods | Caltech-2V | | Caltech-3V | | Caltech-4V | | Caltech-5V | | CCV | |
|---|---|---|---|---|---|---|---|---|---|---|
| | ACC | NMI | ACC | NMI | ACC | NMI | ACC | NMI | ACC | NMI |
| KM | 41.6 | 30.5 | 46.3 | 31.3 | 54.6 | 46.7 | 57.4 | 49.1 | 19.4 | 17.6 |
| Ncuts | 39.9 | 31.2 | 42.6 | 25.4 | 67.8 | 47.6 | 74.1 | 58.0 | 21.6 | 17.8 |
| AvKM | 46.4 | 31.4 | 46.9 | 31.5 | 44.9 | 30.6 | 45.1 | 30.2 | 12.4 | 5.6 |
| AvNcuts | 42.8 | 25.2 | 43.7 | 25.5 | 41.8 | 24.9 | 43.3 | 25.4 | 18.5 | 12.3 |
| CoregMVSC (2011) | 49.2 | 39.6 | 54.4 | 45.3 | 64.9 | 54.5 | 71.2 | 61.9 | 19.0 | 17.2 |
| RMKMC (2013) | 51.4 | 33.5 | 59.5 | 49.4 | 65.5 | 60.3 | 71.1 | 61.9 | 20.5 | 19.0 |
| SwMC (2017) | 34.2 | 26.6 | 30.2 | 23.1 | 43.7 | 44.2 | 34.3 | 34.5 | 10.7 | 0.4 |
| ONMSC (2020) | 57.4 | 45.6 | 58.2 | 56.8 | 62.3 | 66.1 | 62.4 | 67.4 | 12.2 | 5.8 |
| SMVSC (2021) | 49.9 | 37.1 | 54.8 | 43.3 | 60.1 | 53.3 | 68.8 | 64.6 | 20.6 | 16.3 |
| MvSCN (2019) | 45.0 | 35.0 | 67.7 | 61.3 | 56.6 | 64.3 | 77.4 | 78.0 | 11.1 | 4.4 |
| EAMC (2020) | 41.9 | 25.6 | 38.9 | 21.4 | 29.6 | 16.5 | 31.8 | 17.3 | 26.3 | 26.7 |
| MVC-VAE (2020) | 39.9 | 28.1 | 70.8 | 58.5 | 71.4 | 61.7 | 60.6 | 56.0 | 10.4 | 0.04 |
| DEMC (2021) | 39.4 | 22.2 | 38.7 | 27.0 | 48.4 | 39.7 | 47.6 | 35.9 | 12.5 | 7.4 |
| SiMVC (2021) | 50.8 | 47.1 | 56.9 | 50.4 | 61.9 | 53.6 | 71.9 | 67.7 | 14.1 | 11.6 |
| CoMVC (2021) | 46.6 | 42.6 | 54.1 | 50.4 | 56.8 | 56.8 | 70.0 | 68.7 | 29.5 | 28.7 |
| MFLVC (2022) | 60.6 | 52.8 | 63.1 | 56.6 | 73.3 | 65.2 | 80.4 | 70.3 | 28.1 | 30.1 |
| SPDMC (2023) | 46.9 | 34.0 | 51.4 | 40.5 | 66.5 | 60.9 | 86.0 | 77.6 | 10.8 | 6.0 |
| DIVIDE (2024) | 58.2 | 52.9 | 60.9 | 53.8 | 64.3 | 57.9 | 64.6 | 60.4 | 16.4 | 11.1 |
| **LUCE-CMC** | **64.7** | **53.1** | **75.6** | **63.4** | **80.6** | **70.0** | **88.6** | **82.2** | **34.3** | **32.4** |

(All-view Normalized Cuts)), five traditional multi-view clustering methods (**CoregMVSC** [15], **RMKMC** [1], **SwMC** [19], **ONMSC** [36], **SMVSC** [21]), and nine deep multi-view clustering methods (**MvSCN** [13], **EAMC** [35], **MVC-VAE** [34], **DEMC** [27], **SiMVC** and **CoMVC** [22], **MFLVC** [28], **SPDMC** [2], **DIVIDE** [18]).

## 4.4 Implementation Details

Our experimental platform operates on the Windows 10 operating system, leveraging a robust configuration featuring 24GB of system memory and a high-performance NVIDIA GeForce RTX 3090 GPU. The hyperparameters $\tau_1$ and $\tau_2$ have been meticulously configured with values of 0.1 and 1.0, respectively. We ran 20 runs on all datasets, and the proposed models all reached convergence after 100 epochs. In order to mitigate the risk of the proposed method encountering local minima during the training process, a strategic approach was employed. The reported clustering results were based on the attainment of the lowest clustering loss, serving as a safeguard against convergence to suboptimal solutions [22, 35]. Our approach includes a transformer module with a multi-head attention mechanism featuring eight heads. We implemented the model using PyTorch and optimized it with the Adam optimizer, using default parameters and a learning rate of 0.001. The architecture consists of three fully connected layers, each with a ReLU activation function. The output dimensions of the layers are 512, 512, and 256, respectively.

## 4.5 Clustering Performance Analysis

Table 2 presents the clustering efficacy of the LUCE-CMC method across five publicly accessible datasets. For the Caltech dataset,

we generally believe that as the view increases, the clustering effect should get better. This improvement is because the amount of information about the target object has increased, which will theoretically help the clustering process. In short, the more views we look at things, the more comprehensive information we obtain, and the accuracy of clustering will naturally increases. Nevertheless, such an expected improvement in clustering performance is conspicuously absent when observing the outcomes of the SwMC and DEMC methods within the Caltech-5V dataset, in contrast to the Caltech-4V dataset. This discrepancy underscores the dataset's potential in more efficaciously validating the robustness of the clustering method as the number of viewpoints escalates.

Furthermore, the LUCE-CMC method's clustering performance surpasses that of the MVC methodologies reported and compared. This superiority is indicative of the LUCE-CMC method's ability to exploit the enhanced feature-level contrastive learning mechanism for extracting, to the greatest extent possible, feature representations between views that benefit downstream clustering tasks. Additionally, the enhanced cluster-level contrastive learning mechanism illustrates the effectiveness of contrastive fusion features and demonstrates its ability to collect high-quality, multi-view semantic information. This clustering result highlights the critical role that the LUCE-CMC approach plays in improving the clustering performance through the synergistic application of contrastive learning concepts to multi-view data processing.

## 4.6 T-SNE Visualization Analysis

Figure 2 illustrates the superior clustering efficacy of our proposed methodology through a comparative analysis with the optimal outcomes derived from a total of 18 baseline methodologies, notably

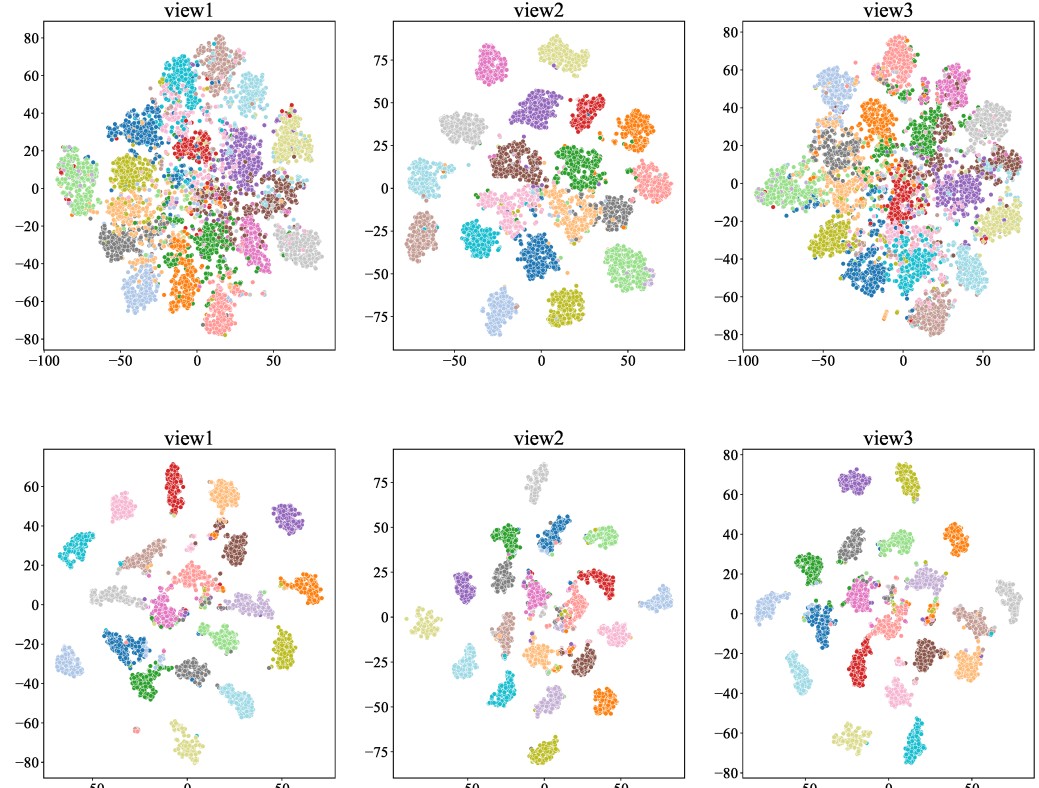

**Figure 2: The visualization of features learned by the proposed method in the CCV dataset. The upper is CoMVC (second best) method and the lower is LUCE-CMC method.**

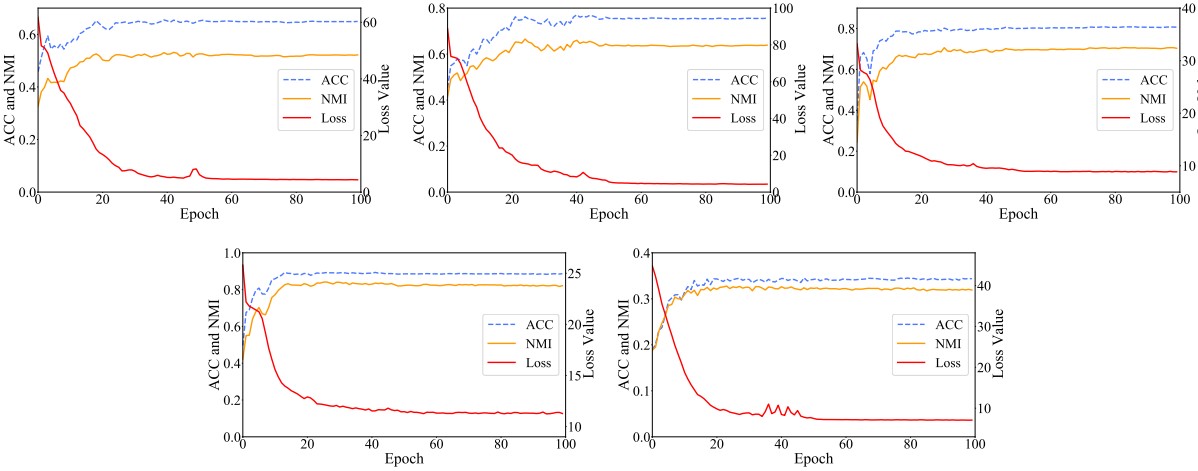

**Figure 3: The convergence curves on multi-view datasets.**

CoMVC, alongside the LUCE-CMC, within the CCV dataset. This visualization elucidates that the LUCE-CMC technique not only augments the intra-cluster compactness, thereby enhancing homogeneity within clusters, but also significantly amplifies the inter-cluster separation, thus ensuring a distinct demarcation between

**Table 3: Ablation study on the multi-view datasets.**

| $\mathcal{L}_{DDC}$ | $\mathcal{L}_{En-FeaCL}$ | $\mathcal{L}_{En-CluCL}$ | Caltech-2V | | Caltech-3V | | Caltech-4V | | Caltech-5V | | CCV | |
|:---:|:---:|:---:|:---:|:---:|:---:|:---:|:---:|:---:|:---:|:---:|:---:|:---:|
| | | | ACC | NMI | ACC | NMI | ACC | NMI | ACC | NMI | ACC | NMI |
| √ | | | 28.6 | 18.6 | 32.4 | 27.7 | 30.5 | 20.1 | 32.7 | 24.6 | 12.6 | 9.2 |
| √ | √ | | 48.9 | 45.1 | 64.8 | 51.0 | 48.9 | 44.1 | 63.1 | 55.1 | 17.7 | 14.0 |
| √ | | √ | 63.2 | 51.7 | 65.0 | 60.0 | 77.1 | 66.0 | 75.2 | 67.3 | 29.9 | 28.3 |
| √ | √ | √ | **64.7** | **53.1** | **75.6** | **63.4** | **80.6** | **70.0** | **88.6** | **82.2** | **34.3** | **32.4** |

dissimilar clusters. This dual enhancement mechanism facilitated by the LUCE-CMC approach underscores its robustness in achieving a more precise and efficient clustering performance.

## 4.7 Ablation Study

Table 3 substantiates the efficacy of individual components within the LUCE-CMC. Initially, employing solely the DDC clustering module yields limited success in fully exploiting the feature representations across different views. This limitation is attributed to the DDC module's intrinsic focus on single-view clustering dynamics, which does not fully leverage the comprehensive and diverse perspectives offered by multi-view data. However, the subsequent integration of the En-FeaCL and En-CluCL modules marks a significant enhancement in clustering performance. The holistic optimization of these components culminates in a substantial improvement in clustering efficacy.

## 4.8 Convergence Analysis

Figure 3 describes the convergence trajectories for the overall loss function, ACC and NMI metrics across diverse dataset configurations, including Caltech-2V, Caltech-3V, Caltech-4V, Caltech-5V, and CCV dataset. The graphical representation clearly demonstrates that the convergence for all three metrics stabilizes post the 60-epoch threshold. This observation underscores the efficacy of the algorithmic iterations in reaching a balancing state, thereby indicating a robust model training process that ensures consistency in performance metrics across multiple dataset variations.

## 4.9 Parameter Analysis

To examine the sensitivity of the hyperparameter $\alpha$, Figure 4 presents a detailed analysis through a curve progression illustration on the typical Caltech-5V and CCV dataset due to space limit. Notably, the curve elucidates a distinct apex, signifying the maximum value point of hyperparameter $\alpha$. This critical juncture is indicative of the optimal balance between various components of the model, harmonizing their contributions to the overall clustering performance. Consequently, the hyperparameter value corresponding to this pinnacle is selected as the optimal choice for our model. This selection process exemplifies a data-driven approach to hyperparameter optimization as existing deep multi-view clustering methods do. Additionally, it is clearly seen that for the two datasets it is simple to select a satisfactory result with a wide range of hyperparameters, i.e., [0.1, 0.4].

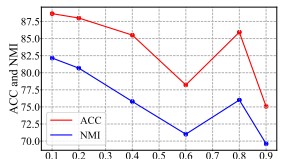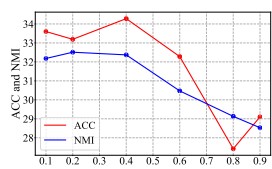

**Figure 4: Clustering performance on ACC and NMI values of the proposed LUCE-CMC method with different values of hyperparameter $\alpha$, i.e.,** $\{0.1, 0.2, 0.4, 0.6, 0.8, 0.9\}$**, on the typical multi-view Caltech-5V and CCV dataset.**

## 5 CONCLUSION

In this paper, we introduce a novel approach by Learning dUal enhanCed rEpresentation for Contrastive Multi-view Clustering (LUCE-CMC), which effectively explores and mines potential information between multi-view data that is inherently beneficial for clustering tasks. Our proposed model includes feature-level with enhanced latent feature representation via feature alignment and reconstitution, and cluster-level with enhanced clustering structure via shared representation. First, we note that feature-level CL is constrained by noise in data, leading to inadequate mining of potential features affecting the feature space. The newly-designed En-FeaCL aims to significantly improve feature representation quality, aligning with the objectives of effective clustering. Second, we introduce a new enhanced cluster-level CL model, leveraging high-level feature fusion to accurately identify similarities and differences among data points across views. By exploiting the above data relationships, LUCE-CMC method achieves refined and effective clustering performance. Extensive experiments show that LUCE-CMC significantly outperforms existing traditional and deep MVC methods. These results highlight our framework's potential into more multi-modal applications.

In the future work, we plan to focus on more challenging contrastive multi-view clustering problem, particularly the partial (i.e., missing or damaged data samples across different views), unpaired (unaligned data samples in multiple views), or trusted (reliable clustering results with different trustworthy scores) contrastive multi-view clustering. These problems raised in various practical scenarios in more recent years, and play important roles in the multi-view/modal processing community. Additionally, we will attempt to address more practical applications, such as multi-modal medical analysis, multi-modal human action recognition, multi-view visual analysis and cross-media sentimental analysis.

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
