# OpenReview forum: "Learning Dual Enhanced Representation for Contrastive Multi-view Clustering"
_acmmm.org/ACMMM/2024/Conference — MM2024 Poster_

### Official Review · Reviewer_KGnm · 2024-05-21

**Rating:** 4
**Confidence:** 4

**Summary:**

A new dual enhanced representation learning scheme (LUCE-CMC) in this paper is designed for quantifying how to enhance feature representations learned at both the feature-level and cluster-level contrastive learning, ultimately benefiting data clustering. In the first component, the authors apply a reconstruction approach to maximize the potential of learned features from each perspective and to minimize the impact of noise in the data. Additionally, in the second component, the authors utilize cluster assignments of fused shared features to mitigate single-view bias.

**Strengths:**

1. The dual enhanced representation learning method introduced in this article represents a significant innovation, outperforming existing multi-view clustering methods. The authors provide a clear explanation of the proposed method, enhancing the reader's understanding of its importance and practical applications.
2. The paper presents a clear motivation and its effectiveness is verified through experiments on multiple datasets, yielding convincing results.
3. This study highlights the significant benefits of applying LUCE-CMC to multi-view datasets, with comprehensive experimental analysis underscoring the method's superiority. The paper is well-structured, clear, and reader-friendly.
4. The article is well-structured with rigorous argumentation, offering valuable insights and references for related research fields.

**Limitations:**

1. In the proposed method section, the authors are suggested to describe the weight fusion part in detail.
2. Authors are requested to standardize the paper format: Ensure inputs and parameters in Algorithm 1 are left-aligned. Formula 9 should not begin with a period, and should follow with uppercase characters. Additionally, the description for Table 1 should end with a period.
3. What is the relevance of discussing the information bottleneck theory in Section 3.7 of this article?

**Suitability:**

3

---

### Official Review · Reviewer_MeFh · 2024-05-21

**Rating:** 5
**Confidence:** 4

**Summary:**

This paper proposes a novel dual enhanced representation method for contrastive multi-view clustering, effectively addressing the feature-level and cluster-level limitations of existing methods. The dual enhanced representation comprises En-FeaCL and En-CluCL. En-FeaCL enhances latent feature representation via feature alignment and reconstitution, while En-CluCL achieves an enhanced clustering structure through the utilization of a shared representation. The innovative method introduced in this paper effectively explores the potential of the feature space, enhances the diversity of view information, and improves clustering performance.

**Strengths:**

The results of the experimental part of the paper can effectively prove that the proposed LUCE-CMC method offers an effective solution for handling complex relationships and information in multi-view data. The advantages are:
1. This article introduces a novel contrastive MVC method grounded in robust theory and practical applicability. The experiments are well-designed, substantiating the method's feasibility and demonstrating scientific discipline.
2. The paper has a clear structure and logical content organization, enhancing reader comprehension. The research method is theoretically sound and validated, with experimental results highlighting its application potential.
3. The experimental comparisons are thorough and inclusive, covering classic clustering, traditional MVC, deep MVC, and contrastive MVC methods, lending greater credibility to the results.

**Limitations:**

The paper also has some limitations, which are:

1. To enhance the paper's quality, it is advisable to cite additional contrastive multi-view clustering references that reflect recent research developments.

2. This article highlights the significant advantages of the proposed method; however, further analysis of its limitations is recommended.

3. The author should detail the mechanism within the En-CluCL section to enhance clarity and readability.

**Suitability:**

3

---

### Official Review · Reviewer_FX5Y · 2024-05-24

**Rating:** 5
**Confidence:** 4

**Summary:**

In this paper, the authors address the problem of contrastive multi-view clustering by proposing a method to learn dual enhanced representations that effectively tackle current challenges. Their approach consists of two main components: enhanced feature-level contrastive learning (En-FeaCL) and enhanced cluster-level contrastive learning (En-CluCL). These components operate alternately, mutually enhancing each other to improve overall clustering performance. Experiments conducted on image and video datasets, including parameter analysis, convergence analysis, and visualization analysis, demonstrate the various strengths of the proposed method.

**Strengths:**

1) The paper is easy to read and understand, with clear connections between different sections.
2) The problem addressed in this paper is significant, and the method has the potential to make a substantial impact on the AI community.

**Limitations:**

1) What are the real differences between the proposed method and existing contrastive multi-view clustering methods? The authors are advised to provide a clear description.
2) The improvement shown in Table 2 is not clearly demonstrated. The authors should present this more clearly.
3) The shortcomings of the proposed method are not analyzed in the Conclusion section.
4) Citing more recent papers would enhance the quality of the paper.

**Suitability:**

3

---

### Official Review · Reviewer_wAEa · 2024-05-24

**Rating:** 4
**Confidence:** 4

**Summary:**

This paper proposes a novel method for contrastive multi-view clustering problem，which combines the enhanced feature-level and cluster-level representation learning modules. The obtained training methods with different parts are mutually reinforcing and beneficial, leading to an improved clustering performance. Lots of resulting Acc and NMI values show the superiority of the proposed method.

**Strengths:**

This article possesses a well-organized framework. The introduction, methods and experimental sections of the article are articulated clearly. The author employs precise professional terminology and the diagrams effectively support the paper's content, and then comprehensively analyzes the shortcomings of existing methods, proposes effective solutions. This paper is well written and the organization is quite clear. The main parts of the proposed method are also described in a very smooth way. The novelty of the method is quite enough, and the focused problem is well addressed.

**Limitations:**

The time complexity can effectively reflect the operational efficiency of the method. It is recommended that the author add a time complexity analysis of the proposed method. Also, a small issue that the text in the figures is too small should be fixed. In the experiments, where do you get the codes of the compared methods? Do you reproduce yourself or get them from the authors? The fair comparison in the experiments is very important.

**Suitability:**

3

---

### Meta-Review · Area_Chair_xHRm · 2024-07-02

**Recommendation:** Accept (Poster)
**Confidence:** 5

**Metareview:**

This paper receives four positive acceptance (3 accept, weak accept). After the rebuttual, two BA are satisfied with the response and raise their scores. All of the reviewers think this paper is well-written and easy to read, with comprehensive evaluation on multiple datasets. Based on this, I vote for accept(Poster) as suggestion.